# Extracellular Heparan 6-*O*-Endosulfatases SULF1 and SULF2 in Head and Neck Squamous Cell Carcinoma and Other Malignancies

**DOI:** 10.3390/cancers14225553

**Published:** 2022-11-11

**Authors:** Yang Yang, Jaeil Ahn, Nathan J. Edwards, Julius Benicky, Aaron M. Rozeboom, Bruce Davidson, Christina Karamboulas, Kevin C. J. Nixon, Laurie Ailles, Radoslav Goldman

**Affiliations:** 1Department of Biochemistry and Molecular & Cell Biology, Georgetown University, Washington, DC 20057, USA; 2Department of Biostatistics, Bioinformatics and Biomathematics, Georgetown University, Washington, DC 20057, USA; 3Clinical and Translational Glycoscience Research Center, Georgetown University, Washington, DC 20057, USA; 4Department of Oncology, Lombardi Comprehensive Cancer Center, Georgetown University, Washington, DC 20057, USA; 5Department of Otolaryngology-Head and Neck Surgery, MedStar Georgetown University Hospital, Washington, DC 20057, USA; 6Princess Margaret Cancer Centre, University Health Network, Toronto, ON M5G 2C1, Canada; 7Department of Medical Biophysics, University of Toronto, Toronto, ON M5G 1L7, Canada

**Keywords:** heparan sulfate 6-*O*-endosulfatase, SULF1, SULF2, head and neck squamous cell carcinoma, cancer associated fibroblast

## Abstract

**Simple Summary:**

6-*O*-endosulfatases, SULF1 and SULF2, are oncogenic in multiple malignancies and are associated with poor survival outcomes. SULF1 is one of the most consistently unregulated enzymes in HNSCC tissues even though its expression in the cancer cells is marginal. Our PDX and RNA scope experiments confirm that SULF1 is provided to the tissues by cancer-associated fibroblasts as opposed to SULF2 supplied by the cancer cells. This paradigm is common to multiple malignancies and suggests a potential for diagnostic and therapeutic targeting of the heparin sulfatases in cancer diseases.

**Abstract:**

Pan-cancer analysis of TCGA and CPTAC (proteomics) data shows that SULF1 and SULF2 are oncogenic in a number of human malignancies and associated with poor survival outcomes. Our studies document a consistent upregulation of SULF1 and SULF2 in HNSC which is associated with poor survival outcomes. These heparan sulfate editing enzymes were considered largely functional redundant but single-cell RNAseq (scRNAseq) shows that SULF1 is secreted by cancer-associated fibroblasts in contrast to the SULF2 derived from tumor cells. Our RNAScope and patient-derived xenograft (PDX) analysis of the HNSC tissues fully confirm the stromal source of SULF1 and explain the uniform impact of this enzyme on the biology of multiple malignancies. In summary, SULF2 expression increases in multiple malignancies but less consistently than SULF1, which uniformly increases in the tumor tissues and negatively impacts survival in several types of cancer even though its expression in cancer cells is low. This paradigm is common to multiple malignancies and suggests a potential for diagnostic and therapeutic targeting of the heparan sulfatases in cancer diseases.

## 1. Introduction

Eukaryotic sulfatases are primarily lysosomal enzymes that hydrolyze sulfate esters during the degradation of macromolecules, often in conjunction with glycosidases [1]. The sulfatase family of proteins shares a common catalytic mechanism using a formyl glycine residue in the active site for catalysis [2]. However, human 6-*O*-endosulfatases SULF1 and SULF2 are distinct from all other sulfatases in that they are neutral pH extracellular enzymes that edit the sulfation of heparan sulfate proteoglycans (HSPGs) instead of degrading them [3]. The HSPG family of core proteins carries one or several serine/threonine residues covalently attached to a heparan sulfate glycosaminoglycan chain further modified by *N*-deacetylation and sulfation, epimerization, and variable *O*-sulfation [4]. Four sulfation sites at the *N*-, 3-*O*-, and 6-*O*-positions of the glucosamine and at the 2-*O*-position of the glucuronic acid [4] regulate protein interactions and the 6-*O*-sulfate-dependent interactions are critical regulators of the pathophysiology of multicellular organisms [5,6,7,8]. The SULF1 and SULF2 enzymes are the only post-synthetic editors of the 6-*O*-sulfation at the internal glucosamines of highly sulfated HSPG domains and their activity defines many critical interactions at the cell surface and in the extracellular matrix (ECM).

HSPGs exquisitely regulate embryogenesis, organogenesis, and physiology of nearly all organs by adjusting gradients of at least 600 proteins including growth factors, cytokines, chemokines, proteases, or collagens [6,9]. The highly specific SULF activities liberate sequestered HS-binding proteins that regulate matrix remodeling, immune infiltration, or signaling of the respective cognate receptors [7,10,11]. The determinants of ligand binding are under intense investigation [12,13,14] and systemic rules need to be further elucidated. We know, however, that heparan 6-*O*-sulfation is essential for the binding of many ligands including VEGF, FGF-1, FGF-10, IL8, HGF, Wnt ligands or L- and P- selectins [4,10]. SULFs represent, therefore, an essential regulatory element that controls the HS-dependent developmental and pathophysiological processes including cancer progression [15,16,17].

Human SULF1 and SULF2 are 65% identical and there is no clear difference in the substrate-specificity of the two enzymes [18]. However, the impact of the two enzymes on cancer diseases is distinct. SULF2 is upregulated and oncogenic in various cancers [7] and a recent study documented that anti-SULF2 antibodies prevent tumor growth in a mouse model of cholangiocarcinoma [19]. The reported impact of SULF1 on cancer progression is less consistent. Widespread low expression of the SULF1 transcript is observed in cancer cell lines [20,21] and prior studies suggested a tumor suppressor function of SULF1 in ovarian, breast, and liver cancers [16]. In contrast, increased SULF1 expression is observed in a wide range of human tumors [7,16] and high SULF1 expression is associated with advanced primary tumor status, higher histological grade, and worse survival in urothelial carcinoma [22]. We have shown that SULF1 and SULF2 enzymes increase in tumor tissues of patients with HNSC and that the increase is associated with poor survival [23]. In this study, we, therefore, examined available datasets to find which cancers are affected by the 6-*O*-endosulfatases and we carried out experiments that verify the unifying concepts in their impact on HNSC and other cancers.

## 2. Materials and Methods

### 2.1. Differential Expression of SULF1 and SULF2 mRNAs in 32 TCGA Studies

RNA-seq data and clinical information of 9160 patients enrolled in 32 cancer studies conducted by the Cancer Genome Atlas (TCGA) consortium (portal.gdc.cancer.gov, accessed on 26 February 2022) and corresponding non-disease tissues from the Genotype-Tissue Expression (GTEx) project (gtexportal.org/home, accessed on 26 February 2021) were downloaded from UCSC-Xena on 26 February 2021 (xenabrowser.net/datapages, accessed on 26 February 2021); SULF1 and SULF2 mRNA was quantified as log_2_(RSEM counts+1) in both datasets. For the differential expression analysis, we selected 14 TCGA cancer studies with n > 10 of paired tumor and normal tissues (Table 1) and we used a Wilcoxon rank sum test to compare paired tumor and non-tumor tissues, where the log_2_-fold change (|Log_2_FC|) > 1 and the false discovery rate (FDR) < 0.05 across studies were considered as statistically significant. SULF1 and SULF2 mRNA expression was further compared by Wilcoxon rank sum test between unpaired tumor tissues of 24 cancer studies and corresponding non-disease tissues of the same organs reported in GTEx (Appendix A).

### 2.2. Differential Expression of SULF Proteins in 10 CPTAC Studies

Proteomic data and clinical information of 1247 patients enrolled in 10 cancer studies with matched tumor and adjacent non-tumor tissue pairs conducted by the Clinical Proteomic Tumor Analysis Consortium (CPTAC) were downloaded from the Proteomic Data Commons (proteomic.datacommons.cancer.gov/pdc, accessed on 18 May 2020) and the CPTAC Data Portal [24] on 18 May 2020. Protein abundance, as determined by the CPTAC Common Data Analysis Pipeline [25] quantified the log_2_ ratio of individual proteins to internal control of each study, using only peptides not shared between quantified proteins. We analyzed the differential expression between tumor and paired normal tissues by paired *t*-test; we compared SULF expression at different cancer stages by one-way ANOVA and computed the corresponding FDRs (Table 2).

### 2.3. Pan-Cancer Survival Analysis Based on SULF1 and SULF2 mRNA

The impact of SULF mRNA expression on time-to-event endpoints as a cancer-driven progression-free interval (PFI), defined by the TCGA research network [26], was evaluated by the Kaplan–Meier method in 20 TCGA cancer studies with >100 patients and >40 PFI events (Appendix A). We identified optimal SULF cutoff values yielding the smallest *p*-value from the log-rank test when both SULF low and high groups, in each study, have at least 25% patients. We used the multivariable Cox-proportional hazard model (MCPH) adjusting for age and gender and summarized hazard ratios (HR) with 95% confidence intervals (CI). We used FDR < 0.05 to call statistical significance. We further evaluated the HNSC TCGA dataset for an association between SULF1 or SULF2 mRNA expression and PFI at different pathological stages of the disease. We analyzed separately the early stage (stage I and II, n = 96) and the late stage (stage III and IV, n = 346) tumors. In each subset, we used the log-rank test as above.

Confirmatory survival analysis was completed on HNSC patients (n = 88) enrolled at the Princess Margaret Cancer Centre, the University of Toronto in line with IRB-approved protocols. The majority of the patients have stage IV oral cancer (Appendix A). RNA was extracted from the snap-frozen tumor tissues using a Qiagen RNeasy mini-kit and sample library preparation was carried out using the Illumina TruSeq stranded total RNA sample preparation kit. Sequencing used a 100-cycle paired read protocol and multiplexing to obtain ~75 million reads/sample on a Novaseq S4 flow cell using XP mode. Transcript abundances in transcripts per million (TPM) were generated from trimmed reads using Kallisto (Pachter Lab, v. 0.46.1) (Berkeley, California, CA, USA) and the gencodev33 human transcriptome or a combined human-mouse transcriptome with reads aligning to mouse removed prior to analysis. Gene-level abundances (TPM) were calculated using the Bioconductor package tximport (v. 1.24.0) (https://bioconductor.org/packages/release/bioc/html/tximport.html, accessed on 26 February 2021). Survival analyses for disease-free intervals were performed using the methods described above.

### 2.4. Cell-Specific Expression of SULF1 and SULF2 in HNSC

We evaluated a published HNSC single-cell RNA-seq dataset [27] that profiled transcriptomes of 5578 cells from tumor tissues of 18 HNSC patients and identified the type of individual cells based on copy-number variations, karyotypes and expression signatures. We downloaded the data from the UCSC Cell Browser (cells.ucsc.edu/?ds=head-neck, accessed on 16 May, 2019) and we quantified the SULF1 and SULF2 mRNAs as log_2_(TPM + 1). Each cell type expressing SULF1 or SULF2 was defined as a percentage of cells with non-zero expression values (Appendix A). Differential expression of SULF1 and SULF2 between the cell types was calculated by non-paired *t*-test.

### 2.5. Correlation Studies of the SULF Enzymes with CAFs and Other Proteins

We computed Pearson’s correlations between SULF1 and all proteins/corresponding mRNAs in the CPTAC study of HNSC [28]. To explore the magnitude of the correlations between SULF1 and cancer-associated fibroblasts (CAFs), the averaged Pearson’s correlation between CAF1 proteins (n = 206) and SULF1 was compared with the averaged SULF1 correlation with all proteins (n = 10,073) where *p*-value was obtained by performing 10,000 permutations using a randomly selected subset of proteins (n = 206). We adopted the CAF1 and CAF2 definitions by Puram et al. [27]. In addition, we adopted the CAF definition of a COL11A1-related CAF subset associated with pre-metastatic locally invasive tumors proposed by Anastassiou [29,30] (Appendix A).

### 2.6. RNAscope Analysis of SULF1 and SULF2 in OSCC Tissues

We selected tumor tissues of patients (n = 20) with carcinoma of the oral cavity (OSCC) for the analysis of SULF1 and SULF2 expression based on in situ hybridization. The patients were either node-positive (n = 10) or node-negative (n = 10) and the node-positive group had, in general, poor survival outcomes (Appendix A). FFPE sections (5 µm) of the patient’s tumors were baked at 60 °C, deparaffinized, and dehydrated. The RNAScope assay (RNAscope Multiplex Fluorescent Reagent Kit v2 #323100) was performed according to the manufacturer’s protocol with probes for Sulf1 and Sulf2 (ACD 403581-C3 and ACD 502241) paired with OPALs 650 and 570 (Akoya FP1496001KT and FP1488001KT, respectively). After the final wash, slides were prepped for IHC and incubated for 60 min with anti-panCK antibody (M3515 DAKO), anti-mouse HRP secondary (DAKO K400111-2), OPAL TSA 520 (Akoya # FP1487001KT), and DAPI (Akoya # FP1490).

Slides were scanned at 10× magnification using the Vectra 3.0 Automated Quantitative Pathology Imaging System (Akoya). Whole slide scans were viewed with Phenochart (Akoya) and high-powered images at 20× (resolution of 0.5 μm per pixel) were selected for multispectral image capture. Three to 20 multispectral image regions of interest (ROIs; 669 μm × 500 μm) were captured in the tumor and normal adjacent regions on each slide. A selection of 10–15 representative multispectral images spanning all tissue sections was used to train the inForm software (tissue/cell segmentation and phenotyping tools). All the settings applied to the training images were saved within an algorithm for batch analysis of all the multispectral images for the project. The analysts were blinded to the patient status and all the raw data were consolidated in PhenoptrReports (Akoya). We quantified the total number of cells in the panCK+ tumor area and the adjacent panCK-area (stroma); these compartments were analyzed for the SULF1+ and SULF2+ cells (Appendix A).

### 2.7. Expression of SULF1 and SULF2 in a PDX Model of OSCC

PDX models were generated as described [31]. RNAseq was carried out on snap-frozen tissues from early passage (passage 1 to 3) PDX models, as described above. Expression of SULF1 and SULF2 was compared between patient and xenograft samples using a paired Wilcoxon rank sum test and visualized using the R package ggplot2 (v. 3.3.6) (RStudio, Auckland, New Zealand).

## 3. Results

### 3.1. SULF1 and SULF2 mRNA Expression in Different Cancer Types

We extended our analysis of HNSC [23] by retrieving the RNA-seq data of 9160 patients in 32 cancer studies from the TCGA database to develop a systematic evaluation of SULF1 and SULF2 expression. We compared tumor and paired normal tissues in 14 cancer studies with >10 available paired samples (Table 1). SULF1 is significantly upregulated in tumor tissues of 10 of the 14 studies of which 9 show >2-fold upregulation. The highest SULF1 log_2_FC is observed in LUAD (log_2_FC = 2.78, FDR < 0.001). SULF2 is overexpressed in 8 cancer types of which 5 increase > 2-fold. The highest SULF2 FC is observed in ESCA (log_2_FC = 2.76, FDR < 0.001). Besides the wide-scale overexpression in tumor tissues, SULF1 is significantly downregulated in KICH and THCA and SULF2 in PRAD but none of the studies reaches a 2-fold decrease.

Our comparison of SULF mRNA in 32 TCGA tumors with the corresponding non-disease tissues from GTEx (Appendix A) shows that SULF1 is significantly increased in 18 cancers of which 16 show >2-fold upregulation. SULF2 is significantly higher in 16 cancer studies, of which 11 show >2-fold increase. All the TCGA studies with significant SULF1 or SULF2 upregulation in paired tissues retain the trend in our GTEx analysis. However, the slight decreases observed for SULF1 in KICH and THCA or for SULF2 in PRAD (Table 1) are not confirmed in the GTEX comparison. The decrease in SULF1 is not significant for any of the GTEX comparisons; SULF2 expression is significantly lower in OV (log_2_FC = −2.27, FDR < 0.001) and UCEC (log_2_FC = −1.76, FDR < 0.001).

### 3.2. SULF1 and SULF2 Proteins in 10 CPTAC Studies

We analyzed the differential expression of SULF proteins in paired tumor and normal tissues of 10 proteomic studies from the CPTAC consortium (Table 2). Seven of the cancer types (LUSC, LUAD, HNSC, KIRC, BRCA, COAD, and LIHC/HCC-HBV) overlap with the TCGA datasets, which enables comparison of the expression at the transcriptional and translational levels. Similar to the pervasive upregulation of SULF1 mRNA, SULF1 protein is significantly upregulated in tumors compared with paired normal tissues in 9 of the 10 studies. Four studies showed >2-fold increase in SULF1 protein in tumor tissues (Table 2) with the highest fold-change observed in HNSC (log_2_FC = 1.59, FDR < 0.001). Only the smaller size (n = 12) study of OSC did not show any difference in SULF1 protein. SULF2 protein is significantly upregulated in the tumors of 6 studies but a > 2-fold increase is only observed in PDAC (log_2_FC = 1.14, FDR < 0.001). In addition, we saw a significant downregulation of SULF2 protein in tumor tissues of HBV-related HCC (log_2_FC = −0.33, FDR < 0.001) and UCEC (log_2_FC = −0.43, FDR = 0.01). This is consistent with the reduced expression of SULF2 mRNA in tumor tissues of LIHC (log_2_FC = −0.481, FDR = 0.116, Table 1) and UCEC (log_2_FC = −1.76, FDR < 0.001, Appendix A). Based on the SULF expression in the two independent datasets (TCGA and CPTAC), we conclude that SULF1 is commonly upregulated across different cancer types while SULF2 overexpression is more restricted to certain cancer pathologies.

We observed >2-fold upregulation of both SULF1 and SULF2 mRNA in four cancer studies from the TCGA (HNSC, ESCA, LUSC, and STAD) and all these cancers remain significantly upregulated compared with the GTEX normal tissues (Figure 1A,B). At the same time, SULF1 and SULF2 proteins are significantly elevated in HNSC and LUSC (Table 1) while the STAD and ESCA studies were not reported at the time of our analysis. Pancreatic cancer (PDAC) is the only CPTAC study with both SULF1 and SULF2 protein significantly upregulated >2-fold in tumors compared with paired normal tissues (Figure 1C). The limited size (n = 4 paired samples) of the PDAC study prevented our analysis of paired mRNA expression in the TCGA dataset. However, SULF1 and SULF2 mRNAs are >20-fold higher in the PDAC tumor tissues than in the non-disease pancreatic tissue from the GTEx (SULF1 log_2_FC = 6.81, SULF2 log_2_FC = 4.85, both *p* < 0.001, Figure 1D), which is consistent with the large difference in the protein expression (Table 2). The survival analyses presented below further support the impact of SULF enzymes on HNSC and PDAC and warrant additional study.

### 3.3. Association of SULF1 and SULF2 mRNA Expression with Survival Outcomes

The association of SULF1 or SULF2 mRNA expression and PFI was analyzed by univariate log-rank tests and compared with published studies (Appendix A). Our literature search found five studies (bladder [22], breast [32], lung [16], gastric [33], and liver [11]) showing that high SULF1 expression is associated with poor survival outcomes. We observed an adverse prognostic trend for these cancers in the TCGA studies but the associations did not reach significance (Appendix A). In addition, high SULF1 is a significant (FDR < 0.05) prognostic factor in KIRP (HR = 2.693), PAAD (HR = 2.365), CESC (HR = 2.227), COAD (HR = 1.867), and LGG (HR = 1.479); we are not aware of studies reporting these associations. We note that SULF1 is significantly increased at the mRNA (Table 1) and protein (Table 2) levels in COAD and negatively impacts survival; such associations deserve further attention. The association of high SULF1 with poor survival (Figure 2) in many cancers is quite remarkable, especially in view of the fact that the SULF1 transcript is low in most cancer cell lines (Appendix A) [20,21].

We found studies of 8 cancers (bladder [22], esophagus [34], head and neck [23], kidney [35], liver [36], lung [37], and pancreatic [38]) showing significant association of SULF2 with survival but the impact is less uniform. Our analysis confirms the reported associations (all FDR < 0.05) of high SULF2 expression with poor PFI of patients with HNSC (HR = 1.687), LIHC (HR = 1.587), and PAAD (HR = 1.724) (Figure 2). An adverse association of SULF2 with PFI in ESCA (HR = 1.352, *p* = 0.158) did not reach statistical significance as reported [34]. In addition, high SULF2 expression is significantly (FDR < 0.05) associated with better PFI in LGG (HR = 0.354, *p* < 0.001) and UCEC (HR = 0.415, *p* = 0.004) (Appendix A) and a favorable prognostic impact of high SULF2 was reported in clear cell renal carcinoma [35] (HR = 0.07, *p* = 0.015, n = 49) and lung squamous cell carcinoma [37] (HR = 0.11, *p* = 0.02, n = 51). These observations were, however, not corroborated in the larger TCGA studies (Appendix A).

### 3.4. Survival Impact of SULF1 and SULF2 in HNSC Differs by Pathological Tumor Stage

We have shown that high SULF1 or SULF2 expression in HNSC is associated with poor survival outcomes [23]. This association was further confirmed in our study of 88 HNSC patients enrolled at the University of Toronto; we used an optimized cutoff of SULF1 (52 high and 36 low expressors) or SULF2 (24 high and 64 low expressors) to show that high expression of either gene is associated with poor disease-free interval (*p* < 0.001) (Appendix A). The study has a limited size but provides an important independent verification of the results.

SULF1 is associated with poor survival in univariate analysis [23] but, contrary to SULF2, loses a significant impact when analyzed in a multivariable model together with SULF2, age, gender, smoking history, tumor stage, and radiation therapy. However, the TCGA sample set used in the analysis is dominated by stage III and IV tumors. To further evaluate the impact of SULF1, we analyzed separately HNSC patients with early (stage I and II) or late (stage III and IV) tumors. We observed that the survival outcomes differ by stage even though SULF1 and SULF2 expression does not differ between early and late-stage tumors [23]. High SULF1 mRNA expression in a tumor is associated with poor PFI in early stage patients (HR = 2.327, *p* = 0.023, Figure 3A) but not in late-stage patients (HR = 1.034, *p* = 0.842). On the contrary, SULF2 mRNA overexpression is significantly associated with poor PFI outcomes in late-stage patients (HR = 1.794, *p* < 0.001, Figure 3B) but not in early stage patients (HR = 1.457, *p* = 0.866). The high impact of SULF1 in the early tumors is even more pronounced in a multivariable model [23] that includes SULF1, SULF2, age, gender, smoking, and radiation therapy. While SULF2 is insignificant in the 97 stage I and II patients (HR = 1.32 (95% CI, 0.58–3.0), *p* = 0.59), high SULF1 remains an independent predictor of poor PFI (HR = 4.61 (1.88–11.3) *p* < 0.001). We speculate that this pattern of adverse prognostic impact is associated with a SULF1 function in the local spread of the disease at an early stage which is complemented by SULF2 activity at later stages of the HNSC progression.

### 3.5. Cell-Specific Expression of SULF Enzymes in HNSC

Analysis of a single-cell RNA-seq dataset of HNSC [27] showed that the percentage of SULF1 and SULF2 positive cells varies substantially across 9 cell types (Appendix A). SULF2 is expressed in 63% of all tumor cells (n = 1389 of 2215) which is the highest representation among all the cell types. The positivity of SULF1 is the highest in fibroblasts (48%, n = 691 of 1440) compared with <20% in any other cell type (Appendix A, Figure 4A). SULF1 is expressed in only 14% of tumor cells (n = 309 of 2215).

SULF1 expression is significantly higher in fibroblasts compared with tumor epithelial cells in terms of both percent positivity and expression; in contrast, SULF2 expression has the opposite trend (Figure 4). The mean SULF1 mRNA, represented as log_2_(TPM+1), is 0.172 in tumor cells compared with 1.099 in fibroblasts (*p* < 0.001); the mean value of SULF2 mRNA is 1.145 in tumor cells compared with 0.386 in fibroblasts (*p* < 0.001) (Figure 4A). The percentage of positive cells among individual patients ranges from 2.4–54.5% for SULF1 and 43.1–91.7% for SULF2 in tumor cells, and from 31.3–71.0% for SULF1 and 8.7–43.8% for SULF2 in fibroblasts (Figure 4B). Interestingly, the mean expression of SULF1 in SULF1-positive cells remains significantly higher in fibroblasts than in tumor cells (2.291 vs. 1.232, *p* < 0.001); however, the mean expression of SULF2 in SULF2-positive fibroblasts and tumor cells is the same (1.826 vs. 1.812, *p* = 0.8, Figure 4C). These results suggest that SULF1 is expressed to a high degree by a large sub-population of fibroblasts; SULF2 is expressed to the same degree in the fibroblasts and tumor cells but the population of tumor cells expressing SULF2 is much bigger than that of the fibroblasts. We conclude that SULF1 and SULF2 in HNSC derive primarily from the fibroblasts and tumor cells, respectively. The results show that the expression of SULF1 and SULF2 in HNSC is regulated by different mechanisms, which leads to an independent regulation of the HS-dependent signaling activities by the two enzymes.

To further strengthen the observation that SULF1 is expressed in fibroblasts and SULF2 in tumor cells, we analyzed the RNA-seq data from the Cancer Cell Line Encyclopedia (CCLE). SULF1 expression in fibroblast cells is distinctly higher compared with all the cancer cell lines (Appendix A) but SULF2 expression is the highest in cell lines from neuroblastoma, HNSC, and other cancers (Appendix A). This suggests that SULF1 expression in fibroblasts of the tumor tissues is not unique for HNSC but is more likely a pan-cancer event.

A final demonstration of the expression of SULF1 in fibroblasts comes from our PDX studies of 42 HNSC patients (Figure 5) showing that in all but one case the expression of SULF1 decreases in the PDX compared with the primary tumor (median primary tumor TPM = 52.6, median PDX TPM = 1.7, *p* < 0.001; Wilcoxon rank-sum test). This is in line with the expansion of tumor cells and loss of the transplanted stroma commonly observed in the PDX models. In contrast, SULF2 expression in the PDX increases (median primary tumor TPM = 50, median PDX TPM = 103, *p* < 0.01; Wilcoxon rank-sum test) which confirms that the tumor cell is the major source of this enzyme.

### 3.6. SULF1 Expression in Cancer-Associated Fibroblasts

Dominant expression of SULF1 in fibroblasts and its increase in HNSC tissues, in spite of low expression in the HNSC cell lines, prompted us to analyze its connection with cancer-associated fibroblasts (CAFs). We examined the correlation of genes (n = 206) of CAF1, an HNSC CAF defined previously [27], with SULF1 in the CPTAC HNSC study. Analysis of Pearson’s correlation coefficients shows that 122 proteins in the CPTAC dataset are correlated with SULF1 with r > 0.55, of which 44 belong to the CAF1 cluster (Appendix A). The distribution of the correlation coefficients of the CAF1 genes is shifted to significantly higher values compared with other proteins (n = 10,073, *p* < 0.001) (Figure 4D). The correlations of the SULF2 protein with the CAF1 genes are weaker, as expected, and we observed a similar trend in the RNAseq data from the CPTAC HNSC study (Appendix A). The expression signature of the HNSC CAF1 [30] overlaps substantially with a COL11A1-expressing CAF which is defined by an invasive pre-metastatic phenotype [29,30]. This subset of CAF was observed in multiple cancers (HNSC, ovarian, pancreatic, colorectal) which strongly suggests that SULF1 is impactful in multiple cancers, in addition to HNSC.

The association of SULF1 with CAFs is further supported by our RNAscope analysis of SULF1 and SULF2 in 20 OSCC patients (Figure 6). The in situ hybridization clearly shows that SULF1-expressing cells are more common in the stroma (mean 24.9% stroma vs. 8.4% tumor, *p* < 0.001) while SULF2 expression is higher in the cancer cells (mean 22.7% stroma vs. 52.5% tumor, *p* < 0.001) (Appendix A). In an exploratory analysis, we separated the OSCC patients into node-positive (n = 10) and node-negative (n = 10) groups and we observed a trend for higher SULF1 and SULF2 expression in the node-positive cases (Figure 6). The results support that SULF1-positive CAFs accumulate at the invasive front of the HNSC at a pre-metastatic stage and facilitate local invasion.

## 4. Discussion

Previous studies showed that SULF1 and SULF2 are upregulated in several cancers [4,16,22,23,34,37,38]. SULF2 is considered oncogenic [7], but the SULF1 impact is more controversial [7,39,40]. Are there unifying trends in the cancer biology of the heparan 6-*O*-endosulfatases supporting their impact on HNSC and other cancers? Proteogenomic analysis of the TCGA and CPTAC datasets conclusively documents a significant elevation of SULF1 and SULF2 in multiple cancer tissues compared with adjacent (Table 1) or normal (Appendix A) counterparts. This is uniformly corroborated by protein increases (Table 2) and at least six cancers (BRCA, HNSC, KIRC, LUSC, LUAD, PAAD) consistently upregulate both SULF1 and SULF2. PDAC mRNA was not reported in the TCGA adjacent normal but both mRNA (compared with normal) and protein show some of the highest increases overall. Other cancers upregulate one of the SULFs (e.g., SULF1 in COAD) or the SULFs remain unchanged, but decreases, such as SULF2 in UCEC, are rare.

At the same time, high expression of SULF1 or SULF2 is typically associated with poor survival (Appendix A). Among all the cancers examined, we observed the most consistent impact of SULF1 and SULF2 in PAAD and HNSC (Figure 1 and Figure 2) but other cancers are affected as well. HNSC is an interesting case because SULF2 negatively affects the survival of stage III and IV patients while the impact of SULF1 is more prominent in stage I and II cancer patients (Figure 3). Associations of high expression with improved outcomes are restricted to SULF2 (e.g., in UCEC) but remain exceptional. High SULF1 predicts significantly lower PFI in at least 5 cancers and other malignancies, reported previously, following a similar trend (Appendix A). The consistent increases of SULF1 in cancer tissues and their uniform association with poor survival outcomes are quite remarkable because SULF1 expression in cancer cell lines is typically low [20,21] (Appendix A). However, our analyses of HNSC (Figure 6) and analyses of other cancers [41] show that SULF1 is high in cancer tissues and supplied by the CAF, a cell type associated with cancer invasion, metastasis, and the escape from immune surveillance [30,42,43] and so far overlooked in the cancer biology of the SULF enzymes.

Our PDX study of HNSC shows that SULF1, contrary to SULF2, disappears in the transplanted tumors, as expected for a gene expressed by the stromal cells (Figure 5). The scRNAseq data [27] show a strong correlation of SULF1 with CAF1 genes (Figure 4) and the genes of the COL11A1-expressing CAFs associated with locally invasive pre-metastatic cancer disease [29] (Appendix A). SULF1 is a gene typical of this subtype of CAF in several cancers [30] which supports that the CAF supply SULF1 not only in HNSC but in general; the function of SULF1 in the biology of the CAF deserves further attention. Finally, our RNAScope study shows that SULF1+ cells localize to the stroma while SULF2+ cells overlap to a large degree with the cytokeratin+ tumor cells (Figure 3). Our results also suggest that SULF1+ cells in the stroma are more abundant in node-positive tumors which is in line with recent papers associating low tumor/stroma ratio [44] or the presence of CAF [45] with poor HNSC survival outcomes.

## 5. Conclusions

Our study confirms the overexpression of SULF1 and SULF2 in various cancers, which is commonly associated with poor survival outcomes; the 6-*O*-endosulfatases emerge as interesting targets for cancer monitoring and therapeutic intervention. It is expected that the enzymes determine the survival of HNSCC patients by adjusting gradients of heparan sulfate binding proteins in the microenvironment of tumors. We have strong evidence that SULF1 is supplied by CAF while SULF2 is provided primarily by the cancer cells, which has important consequences because the secreted SULF enzymes act locally due to strong non-covalent interactions with cell surfaces [7]. In addition, recent data suggest that their activity is regulated by cell-specific posttranslational modifications [46]. We know that SULFs regulate oncogenic pathways but they also adjust matrix structure, angiogenesis, or immune responses [3,4,7,8,10,15,16,17,47,48], and their function at the tumor/stroma interface in different cancers needs further study.

## Figures and Tables

**Figure 1 cancers-14-05553-f001:**
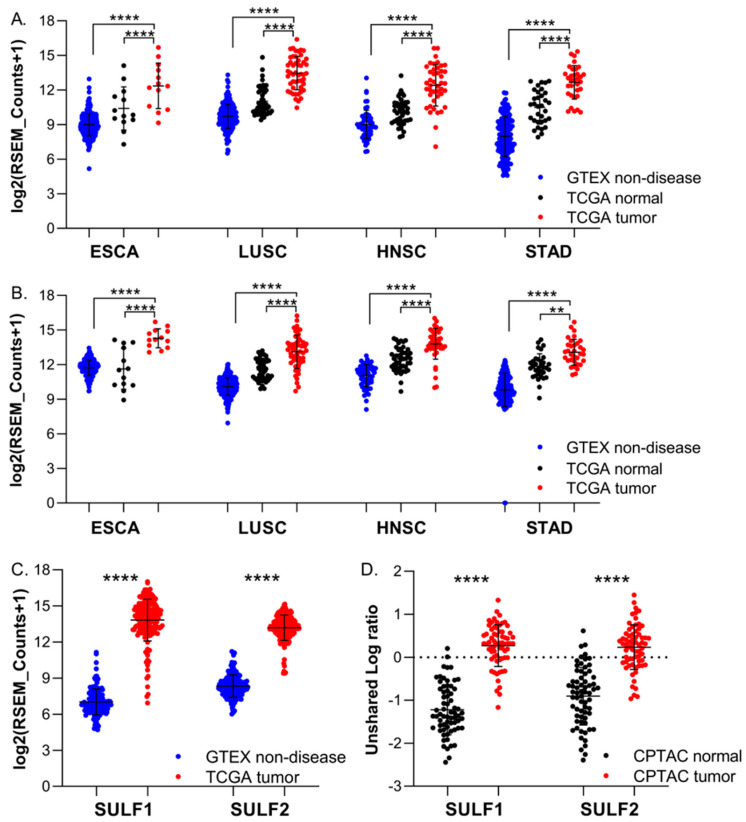
Expression of SULF1 and SULF2 in five cancer types. (**A**) SULF1 and (**B**) SULF2 mRNA in tumor compared with paired normal tissue of cancer patients in TCGA datasets and to non-disease tissues of healthy donors from GTEx; (**C**) SULF1 and SULF2 mRNA in the PAAD tumor tissues from TCGA compared with non-cancerous pancreatic tissue from GTEX. (**D**) SULF1 and SULF2 protein in the PDAC tumor and adjacent non-cancer tissues from CPTAC; *p* < 0.0001 (****), *p* < 0.01 (**).

**Figure 2 cancers-14-05553-f002:**
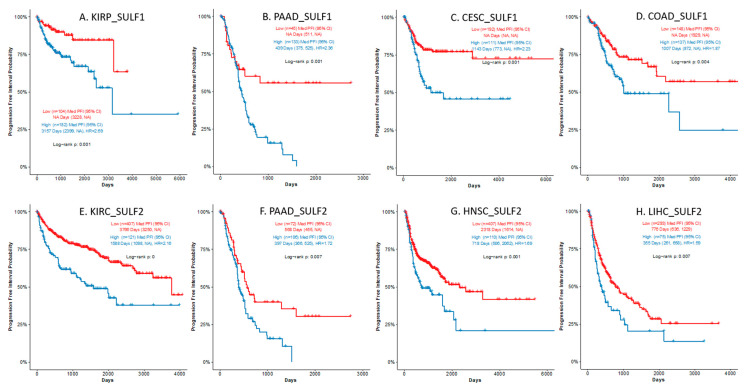
SULF1 and SULF2 expressions in tumor tissues are associated with poor survival. High SULF1 (**A**–**D**) or high SULF2 (**E**–**H**) expression is significantly (HR > 1.5, FDR < 0.05) associated with poor progression-free interval (PFI) in the following TCGA cancer studies: (**A**). KIRP, (**B**). PAAD, (**C**). CESC, (**D**). COAD for SULF1; and (**E**). KIRC, (**F**). PAAD, (**G**). HNSC, and (**H**). LIHC for SULF2.

**Figure 3 cancers-14-05553-f003:**
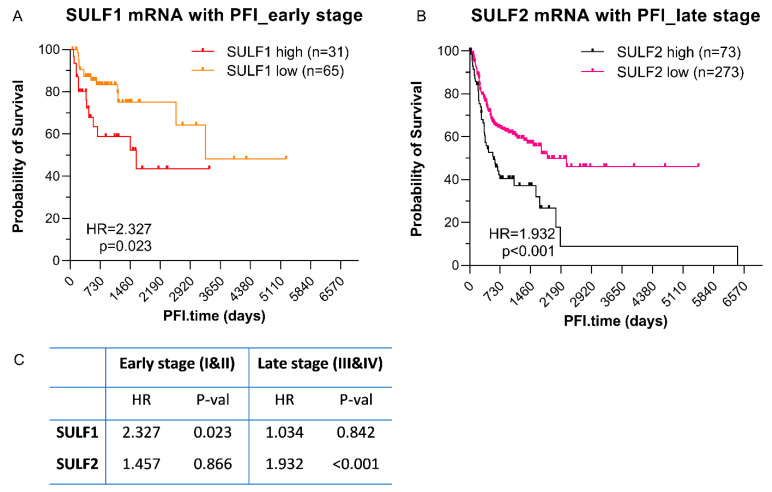
Impact of SULF1 and SULF2 expression on PFI of HNSC patients differs between early and late-stage tumors. (**A**) SULF1 is associated with PFI in early stage HNSC; (**B**) SULF2 is associated with PFI in late-stage HNSC. (**C**) Summary statistics of the PFI in the early and late-stage tumors.

**Figure 4 cancers-14-05553-f004:**
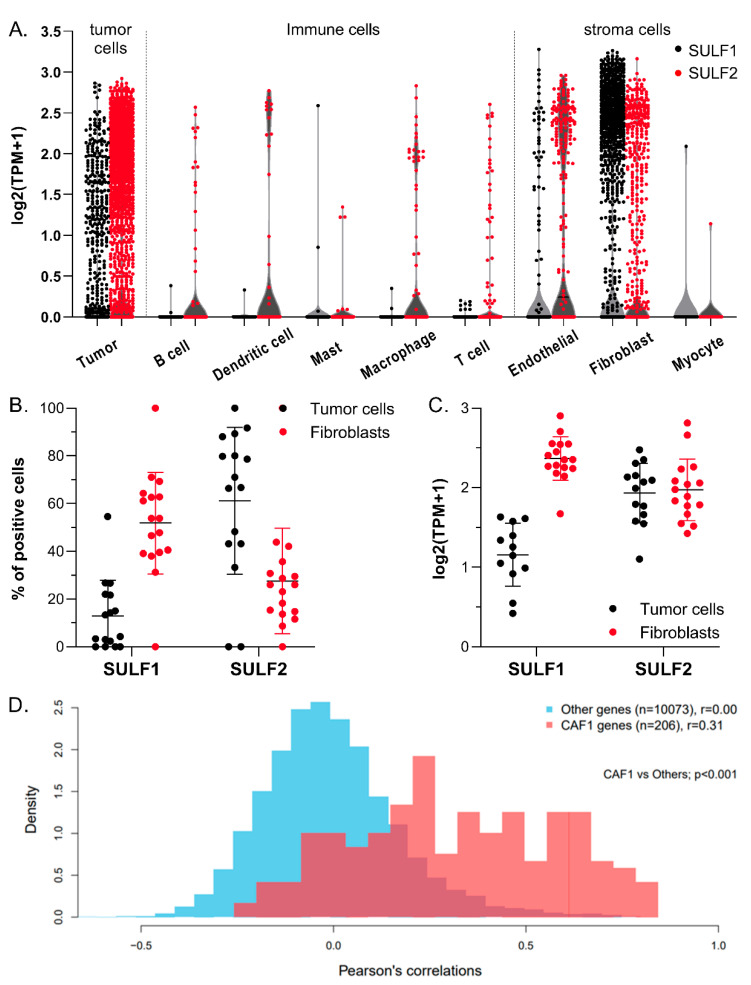
SULF1 and SULF2 are differentially expressed in HNSC epithelial and fibroblast cells. (**A**) SULF1 and SULF2 mRNA expression in different cell types detected in HNSC tumors; (**B**) percent of cells expressing SULF1 or SULF2 differ between the tumor epithelial and fibroblast cells (*p* < 0.0001, two-way ANOVA); (**C**) expression of SULF1 mRNA in SULF-positive cells differs between tumor epithelial cells and fibroblast (*p* < 0.001) but SULF2 does not (*p* = 0.800); and (**D**) distribution of Pearson’s correlation coefficients of SULF1 with CAF1 (n = 206) and other (n = 10,073) proteins measured in the CPTAC HNSC study [28]. Correlation coefficients of SULF1 protein (red bars) with the CAF1 proteins are significantly (*p* < 0.001) higher than the correlations with other proteins. Analyses and the definition of CAF1 are based on a single-cell RNA-seq study of 5578 cells in tumor tissues of 18 HNSC patients [27].

**Figure 5 cancers-14-05553-f005:**
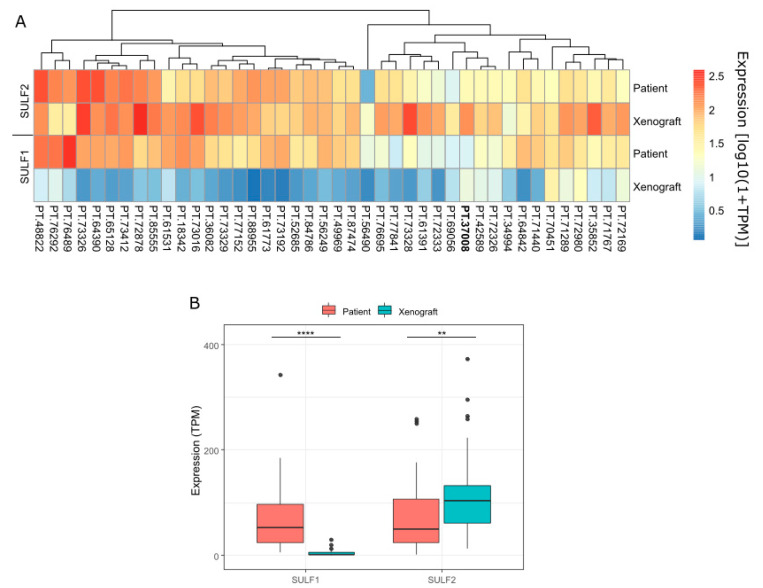
Comparison of RNA expression of SULF1 and SULF2 in primary HNSC tumors (n = 42) and their PDX in immunocompromised mice: (**A**) heatmap of log10(1+TPM) expression of SULF1 and SULF2. Patient ID in bold represents the one case in which SULF1 expression does not decrease from patient to PDX; (**B**) boxplots of SULF1 and SULF2 expression (TPM) in Patient and PDX samples (** *p* < 0.01, **** *p* < 1 × 10^−4^; Wilcoxon rank-sum test).

**Figure 6 cancers-14-05553-f006:**
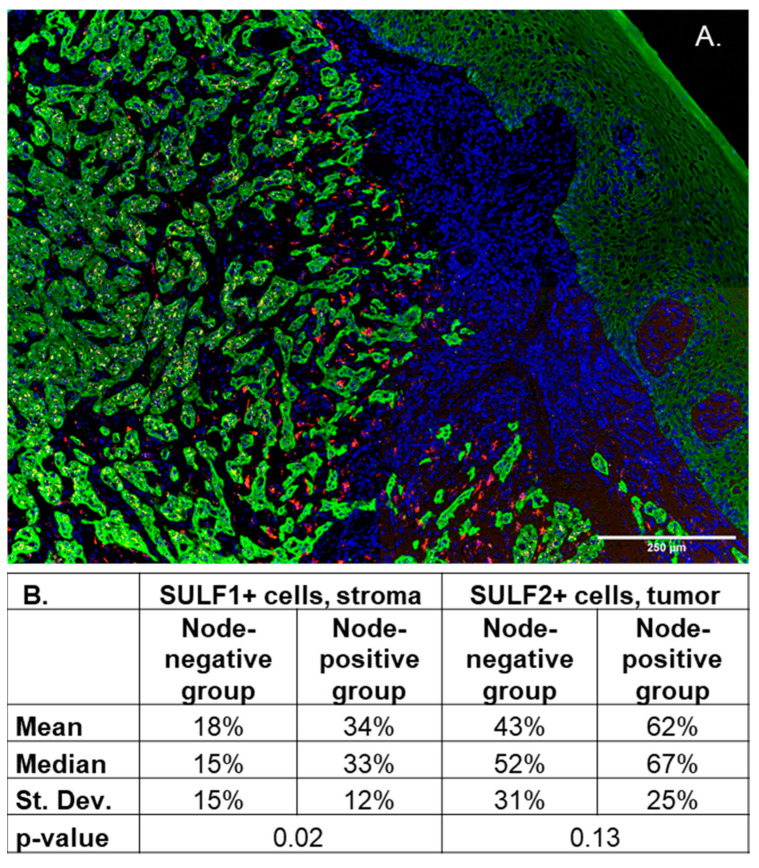
RNAScope of OSCC tumors: (**A**) a tongue cancer stained for DAPI (blue), cytokeratin (green), and with in situ probes for SULF1 (red) and SULF2 (yellow). SULF1 localizes to the stroma, and SULF2 is expressed mainly in the tumor epithelial cells. Adjacent normal tissue is mostly negative for both SULF1 and SULF2; (**B**) comparison of SULF1+ and SULF2+ cell counts in the cancer cell (cytokeratin+) and adjacent stroma (cytokeratin-) of OSCC patients (n = 20). Percentage of SULF1+ cells in the stroma is significantly higher (*p* = 0.023) in node-positive patients (n = 10) with poor survival than in node-negative (n = 10) patients with good survival. SULF2* cells in the tumor follow a similar trend but the difference is not significant.

**Table 1 cancers-14-05553-t001:** Differential expression of SULF1 and SULF2 mRNA between paired tumor and normal tissues in 14 TCGA studies. Entries with fold-change > 2 and FDR < 0.05 are in bold.

TCGA Study			SULF1		SULF2	
Project	Primary Name	No. Pairs	log_2_FC	FDR	log_2_FC	FDR
LUAD	lung adenocarcinoma	58	**2.78**	**5.73 × 10^−12^**	0.74	8.52 × 10^−4^
ESCA	esophageal carcinoma	13	**2.76**	**6.50 × 10^−3^**	**2.65**	**8.52 × 10^−4^**
LUSC	lung squamous cell carcinoma	50	**2.62**	**8.70 × 10^−13^**	**1.58**	**1.31 × 10^−8^**
COAD	colon adenocarcinoma	26	**2.55**	**7.78 × 10^−5^**	0.50	2.12 × 10^−2^
HNSC	head and neck squamous cell carcinoma	43	**2.52**	**5.20 × 10^−7^**	**1.32**	**3.44 × 10^−5^**
STAD	stomach adenocarcinoma	33	**2.46**	**1.48 × 10^−5^**	**1.22**	**8.52 × 10^−4^**
BLCA	bladder urothelial carcinoma	19	**2.14**	**2.80 × 10^−4^**	0.56	2.75 × 10^−1^
BRCA	breast invasive carcinoma	112	**2.03**	**1.08 × 10^−25^**	0.82	1.31 × 10^−8^
KIRC	kidney renal clear cell carcinoma	72	**1.31**	**1.85 × 10^−6^**	0.98	4.47 × 10^−7^
LIHC	liver hepatocellular carcinoma	50	0.90	1.07 × 10^−1^	−0.48	1.16 × 10^−1^
KIRP	kidney renal papillary cell carcinoma	32	0.23	8.75 × 10^−1^	**1.31**	**4.45 × 10^−5^**
PRAD	prostate adenocarcinoma	51	−0.15	1.07 × 10^−1^	−0.75	4.45 × 10^−5^
THCA	thyroid carcinoma	59	−0.51	4.98 × 10^−3^	0.26	4.01 × 10^−2^
KICH	kidney chromophobe	25	−0.60	1.23 × 10^−2^	−0.77	9.36 × 10^−2^

**Table 2 cancers-14-05553-t002:** Differential expression of SULF1 and SULF2 proteins between tumor and adjacent normal tissues in 10 CPTAC studies. Entries with |(log2FC)| > 1 and FDR < 0.05 are in bold. ND, not detected.

CPTAC Study			SULF1		SULF2	
Project	Primary Name	No. Pairs	log_2_FC	FDR	log_2_FC	FDR
HNSC	head and neck squamous cell carcinoma	68	1.59	1.90 × 10^−15^	0.5	5.37 × 10^−7^
BRCA	breast invasive carcinoma	17	1.51	2.80 × 10^−5^	0.87	2.14 × 10^−3^
PDAC	pancreatic ductal adenocarcinoma	66	1.49	5.47 × 10^−18^	1.14	1.45 × 10^−16^
LUSC	lung squamous cell carcinoma	102	1.35	3.82 × 10^−28^	0.45	1.62 × 10^−11^
LUAD	lung adenocarcinoma	100	0.95	5.47 × 10^−18^	0.19	5.37 × 10^−3^
COAD	colon adenocarcinoma	96	0.73	2.16 × 10^−14^	ND	ND
UCEC	uterine corpus endometrial carcinoma	30	0.5	1.69 × 10^−3^	−0.43	1.09 × 10^−2^
KIRC	clear cell renal cell carcinoma	84	0.46	4.16 × 10^−4^	0.26	5.22 × 10^−3^
HBV-HCC	HBV-related hepatocellular carcinoma	160	0.23	5.69 × 10^−2^	−0.33	6.83 × 10^−7^
OSC_JHU	ovarian serous cystadenocarcinoma	12	0.43	7.04 × 10^−2^	0.44	3.80 × 10^−2^
OSC_PNNL	ovarian serous cystadenocarcinoma	10	0.07	4.32 × 10^−1^	−0.58	3.13 × 10^−1^

## Data Availability

The data that support the findings of this study are openly available in UCSC-Xena (https://xenabrowser.net/datapages/, accessed on 26 February 2021) and CPTAC (https://proteomic.datacommons.cancer.gov/pdc/, accessed on 18 May 2020).

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
