# Peer review of "Extracellular Heparan 6-O-Endosulfatases SULF1 and SULF2 in Head and Neck Squamous Cell Carcinoma and Other Malignancies"

_cancers, 2022, doi:10.3390/cancers14225553_

Round 1

Reviewer 1 Report

The current manuscript entitled “Extracellular Heparan 6-O-endosulfatases SULF1 and SULF2 in HNSC and other malignancies” (Yang et al., cancers-1990257) describes their pan-cancer analysis using TCGA and CPTAC data, focusing on SULF’s expression. In TCGA, the expression level of SULF1 and SULF2 was significantly higher in tumor tissues of LUAD, ESCA, LUSC, HNSC, and STAD. In proteomic analysis, the expression was significantly and commonly higher in HNSC. It is likely that the expression of SULFs was cell-type restricted, i.e., SULF1 expression was abundant in cancer stroma whereas SULF2 expression in cancer cells in OSCC tumors (Figure 6). The differential expression is supported by patient-derived xenograft model in HNSC tumors (Figure 5), also differential expression of SULF1 and SULF2 in tumor cells and CAF (Figure 4). 

1) Graphical Abstract
The present study does not include discussion on enzymatic activity of SULFs. Therefore the reviewer wonders if the graphical abstract fits the content of the present study. The authors' precedent paper (ref 23) examined enzymatic function of SULFs by detecting 6-O-sulfation, therefore the reviewer understands that SULFs in HNSC are biologically active.   2) Figure 4.  Ref.27 TableS5 defines SULF1 and SULF2 genes as differentially expressed genes between CAF subsets (CAF1 genes). Therefore the Pearson’s correlations indicated in panel D is reasonable. It is likely that there are three fibroblast populations in the HNSC tissues, i.e., SULF1+SULF2+, SULF1+SULF2-, and SULF1-SULF2-. Are there any phenotypic differences among the three populations?   3) Figure 6.  
It is likely that the tissues have been stained with four colors, which are not distinct in a present form. In addition to the merged image (present one), figures with each color should be indicated. Especially green and yellow are difficult to be recognized. Scale bar should be included in the figure.  SULF1+ cells in tumor and SULF2+ cells in stroma are not shown. The cell numbers are different between the two node-positive and negative groups?   It is interesting if the SULF1+ CAF accumulate at the invasive front of the HNSC tissue, which can facilitate local invasion. Actually a cartoon (Figure 7) in ref 27 suggests involvement of stroma cells in the cancer invasion. Actually was there SULF1-positive area accumulated around the cancer?   Ref 27 comments on the presence of p-EMT in HNSCC tissues. Are there any p-EMT population in the HNSC tissues? SULF1 and SULF2 were present in the p-EMT region of the tissue?
If SULF1+ fibroblasts are accumulated in a restricted region, are there any histochemical changes of surrounding extracellular matrix?
Minor comments.
1) The abbreviated words are recommended not to be included in the title or the abstract. In general, definition of the abbreviations is incomplete.
2) line 29 I contrast —> In contrast?
3) line 372 previouslyl —>  previously
4) The reference numbers are doubled.

Author Response

Point 1: Graphical Abstract. The present study does not include discussion on enzymatic activity of SULFs. Therefore the reviewer wonders if the graphical abstract fits the content of the present study. The authors' precedent paper (ref 23) examined enzymatic function of SULFs by detecting 6-O-sulfation, therefore the reviewer understands that SULFs in HNSC are biologically active.  

Response 1: The graphical abstract is edited according to reviewer 1’s suggestion. The scheme of the protein structure is removed from the graph.

Point 2: Figure 4.  Ref.27 TableS5 defines SULF1 and SULF2 genes as differentially expressed genes between CAF subsets (CAF1 genes). Therefore the Pearson’s correlations indicated in panel D is reasonable. It is likely that there are three fibroblast populations in the HNSC tissues, i.e., SULF1+SULF2+, SULF1+SULF2-, and SULF1-SULF2-. Are there any phenotypic differences among the three populations?  

Response 2: This is an excellent question but we do not know the cell phenotypes (not provided by Puram's scRNAseq study, Ref27). The SULF1+ cells are enriched in the fibroblasts and the SULF2+ cells are enriched in the cancer cells, which means that their phenotypes are different. The SULF1+/SULF2+ cells are few overall.

Point 3: Figure 6. It is likely that the tissues have been stained with four colors, which are not distinct in a present form. In addition to the merged image (present one), figures with each color should be indicated. Especially green and yellow are difficult to be recognized. Scale bar should be included in the figure. SULF1+ cells in tumor and SULF2+ cells in stroma are not shown. The cell numbers are different between the two node-positive and negative groups? It is interesting if the SULF1+ CAF accumulate at the invasive front of the HNSC tissue, which can facilitate local invasion. Actually a cartoon (Figure 7) in ref 27 suggests involvement of stroma cells in the cancer invasion. Actually was there SULF1-positive area accumulated around the cancer? Ref 27 comments on the presence of p-EMT in HNSCC tissues. Are there any p-EMT population in the HNSC tissues? SULF1 and SULF2 were present in the p-EMT region of the tissue? If SULF1+ fibroblasts are accumulated in a restricted region, are there any histochemical changes of surrounding extracellular matrix? 

Response 3: We corrected Figure 6 as suggested and we added Supplementary Figure 3 that shows the three individual channels of the same Figure (PanCK, SULF1, and SULF2). The SULF1+ cells are enriched in the stroma of patients with node positive tumors. SULF1 is expressed very little in the cancer cells and these cells were not analyzed. Similarly, SULF2+ cells are enriched in the tumor and somewhat higher in the patients with node positive tumors. We did not analyze them in the fibroblasts. The SULF1+ cells accumulate at the invasive front but the phenotype is quite heterogeneous and not easy to quantify. A larger study will be needed to address this interesting question. We did not examine p-EMT in the HNSCC tissues.

Point 4: The abbreviated words are recommended not to be included in the title or the abstract. In general, definition of the abbreviations is incomplete.

Response 4: We replaced the abbreviations in following order:1. HNSCC in title is replaced with Head and Neck Squamous Cell Carcinoma; 2. scRNAseq in abstract (line 29) is replaced with single-cell RNAseq; 3. PDX in abstract (line 31) is replaced with patient-derived xenograft.

Point 5: line 29 I contrast —> In contrast?

Response 5: We corrected the text (in line 30 now) as suggested

Point 6: line 372 previouslyl —>  previously

Response 6: We corrected the text as suggested

Point 7: The reference numbers are doubled.

Response 7: We removed the duplicate reference numbers

Reviewer 2 Report

This manuscript substantially bolsters the previous published evidence that SULF1 and SULF2 transcripts are overexpressed in a multiple cancers and in several cases correlate with poor patient outcomes in several cancers.  The finding that high SULF1 correlates with poor PFI in early stage HNSCC while SULF2 correlates with poor PFI is late stage HNSCC is entirely novel and potentially very important for future use of these enzymes for diagnostic/prognostic purposes.   A further novel contribution is that protein expression levels, as determined by quantitative mass spec (CPTAC data) also show significant upregulation in several cancers, with pancreatic cancer showing the largest increase relative to normal tissue.  Prior to this mass spec analysis, evidence for elevation of protein levels largely relied on histochemical studies, which provide only a semiquantitative assessment and are subject to concerns about the specificity of the staining reagents. 

The most striking aspect of the present study is the finding that SULF1 tends to localized to CAFs, whereas SULF2 is most commonly enriched in tumor cells.  As the authors highlight, this finding will encourage mechanistic investigation of how the two SULFs with different localizations each contribute to cancer growth, metastasis and evasion from the immune system.  This discovery should overturn the prevailing view that the two SULFs are largely redundant in tumorigenesis. 

The SULFs in general have not received enough attention from cancer biologists and this study should elevate their visibility and focus attention on the need to study both of them.

I have one request for revision.  In the Simple Summary and Abstract the authors state:  “SULF1 and SULF2, are oncogenic in multiple malignancies including head and neck squamous cell carcinoma (HNSCC) and are associated with poor survival  outcomes.”  It is an overstatement to say that SULF1 and SULF2 have been shown to be oncogenic in HNSCC as the previous studies with this cancer are correlative and do not include direct tests of oncogenic function.    I therefore ask the authors remove the reference to HNSCC in these two summary statements.

Author Response

Point 1:  In the Simple Summary and Abstract the authors state:  “SULF1 and SULF2, are oncogenic in multiple malignancies including head and neck squamous cell carcinoma (HNSCC) and are associated with poor survival  outcomes.”  It is an overstatement to say that SULF1 and SULF2 have been shown to be oncogenic in HNSCC as the previous studies with this cancer are correlative and do not include direct tests of oncogenic function. I therefore ask the authors remove the reference to HNSCC in these two summary statements.

Response 1: The sentence is corrected as suggested: “SULF1 and SULF2 are oncogenic in multiple malignancies and are associated with poor survival outcomes.”